# Studies on Quality Deterioration and Metabolomic Changes in Oysters Induced by Spoilage Bacteria During Chilled Storage

**DOI:** 10.3390/foods14020193

**Published:** 2025-01-09

**Authors:** Hanzheng Dou, Wenxiu Zhu, Siyang Chen, Yue Zou, Xiaodong Xia

**Affiliations:** 1School of Food Science and Technology, Dalian Polytechnic University, Dalian 116034, China; dhz5031067@163.com (H.D.); 18742058310@163.com (W.Z.); csy10_4sym@163.com (S.C.); 1717010103@xy.dlpu.edu.cn (Y.Z.); 2National Engineering Research Center of Seafood, Dalian 116034, China; 3State Key Laboratory of Marine Food Processing and Safety Control, Dalian 116034, China

**Keywords:** spoilage bacteria, metabolomics, oyster, quality deterioration, storage

## Abstract

The correlation between spoilage bacteria and the degradation of aquatic food quality during chilled storage is substantial. However, our understanding of the precise roles of spoilage bacteria in oyster spoilage remains incomplete. The aim of this study was to explore the role of three dominant spoilage bacteria strains in oyster spoilage. Subsequently, the metabolites of spoiled oyster meat after inoculation with bacteria were analyzed using LC-MS-based untargeted metabolomics. Combining the results from various biochemical indicators of spoilage, *Psychrobacter immobilis*, *Shewanella putrefaciens*, and *Photobacterium swingsii* are shown to be the main spoilage bacteria in spoiled oyster meat, and their effects on changes in oyster meat quality were evaluated through total volatile basic nitrogen (TVB-N), pH, thiobarbituric acid reactive substances (TBARSs), and weight loss, respectively. The results showed that *Ps. immobilis* and *S. putrefaciens* exhibited great spoilage capacity. *P swingsii*, although a dominant spoilage bacterium, exhibited lower spoilage competency than the above two bacterial strains but demonstrated activity in producing microbial lipases to oxidize fats. In addition, the results of the metabolomics of spoiled oyster meat suggest that 7, 8-Dimethoxy-3-(4-methoxyphenyl)-4-oxo-4H-chromen-5-yl-2-O-pentopyranosylhexopyranoside, 1,2,3,6-Tetrahydropyridine-4-carboxylic acid, Propionic acid, and L-phenylalanine are potential markers of spoilage in oysters. These findings extend our understanding of the roles that microorganisms play in the spoilage of oysters and offer valuable insights into the development of technologies for monitoring the freshness of oysters based on these potential spoilage markers.

## 1. Introduction

Oysters are a marine shellfish with important economic value and are the most farmed shellfish worldwide. People around the world consume large quantities of oysters, which are an excellent source of protein, trace minerals, and other nutrients [1]. The total production of oysters ranked first among all shellfish production in China. Oysters are a nutritious source of high-quality protein. Oysters have a high water content and are rich in nutrients, and tend to spoil during transport and storage. Biochemical reactions such as enzyme activity, oxidation, and the metabolic activities of specific marine spoilage organisms (*Shewanella putrefaciens*, *Pseudomonas* spp., *Photobacterium* spp., *Aeromonas* spp., etc.) are the main causes of seafood spoilage. These particular spoilage organisms contribute to the deterioration of the sensory characteristics of seafood throughout the storage process, making it unsuitable and unsafe to eat [2,3]. Therefore, understanding the spoilage process is important to maintaining quality and reducing seafood waste.

The spoilage of fresh oysters is a complex phenomenon that typically involves microbial metabolism alongside various biochemical processes, including oxidation and enzymatic activity. The storage circumstances and biochemical makeup will determine the spoilage process. Mollusk shellfish exhibit a unique degradation pattern relative to other seafood owing to their elevated glycogen carbohydrate content. This physiological distinction changes spoilage patterns when glycogen is metabolized into organic acids by bacteria through glycolytic pathway and fermentation. Some studies have used the pH of oyster meat as an indicator of decomposition. Shellfish with a pH value below 5 are usually considered spoiled, while shellfish with a pH value between 6.1 and 5.6 are considered to be transitioning from healthy to acidic [4].

The degradation of aquatic products during storage is primarily indicated by several biochemical markers, including microbial growth, increased volatile saline nitrogen levels, fat oxidation, protein denaturation (indicated by decreased sulfhydryl content, disulfide bond formation, calcium ion release, and ATP enzyme activity), ATP degradation and metabolite production, biogenic amine accumulation, structural changes, and altered heating characteristics [5,6]. Research has shown that there is a strong correlation between microbial activity and seafood spoilage, a phenomenon that has led to extensive studies on the role of microorganisms in this process. For example, Wang et al. [7] examined the ability of *S. putrefaciens* XY07 to cause spoilage in bigeye tuna. Similarly, Liu et al. [8] tested how bacteria isolated from bighead carp could make total volatile basic nitrogen (TVB-N), biogenic amines, and volatile organic compounds (VOCs). Endogenous enzyme activity and oxidation play roles in seafood spoilage; microbial metabolic activity is the main cause of this spoilage. Tan et al. [9] observed that spoilage microorganisms release proteases during cold storage, which cause protein breakdown. In addition, microbial activity leads to the formation of metabolites such as biogenic amines, organic acids, and sulfides [10].

Spoilage and/or pathogenic bacteria in oyster gills during refrigeration have been studied. Nonetheless, there is limited understanding regarding the metabolites generated by spoilage bacteria during the spoilage of oysters. This study aimed to examine the influence of different bacterial metabolic activities on spoilage processes in oysters and to search for the marker metabolites of oyster spoilage and related metabolic pathways using LC-MS-based untargeted metabolomics and biochemical index studies. At the same time, through the aforementioned research methods, the mechanisms of harmful metabolite production in aquatic products during spoilage can also be preliminarily elucidated. Determining specific spoilage bacteria and spoilage potential helps predict the shelf life of aquatic products and adopt effective techniques for their preservation, thereby extending their shelf life.

## 2. Materials and Methods

### 2.1. Sample Preparation and the Analysis of the Microbial Composition

#### 2.1.1. Preparation of Shelled Oyster Meat

Live Pacific oysters (*Crassostrea gigas*) were purchased from Qianhe seafood market (Dalian, Liaoning, China) and subsequently transferred to the laboratory. The oysters were scrubbed under running water and placed in a refrigerator at 4 °C for 2 h to ensure temperature uniformity. Shucking was performed with a knife and the meat was collected and packed in sterile sealed bags and refrigerated at a temperature of 4 °C. All the experiments were performed following the animal ethics guidelines approved by the Animal Ethics Committee of Dalian Polytechnic University (DLPU2024DT008).

#### 2.1.2. Microbiological Analysis of Oyster Spoilage

The bacteria in the meat were gathered using the methodology outlined by Zhuang et al. [11]. The complete procedure of bacterial collection was conducted at a temperature of 4 °C. The DNA of each bacterial sample was separately extracted using a bacterial genomic DNA extraction kit (Omega Bio-Tek, Norcross, GA, USA). Ultimately, bacterial DNA collected from three individual oysters was mixed in equimolar proportions, to compose the final DNA sample of one day.

Microbial composition analysis was conducted on the oyster meat samples collected on day 0, day 3, day 6, and day 10. The 16S rRNA V3-V4 amplicons were amplified using two generic bacterial 16S rRNA gene amplicon PCR primers (PAGE purified). The two common bacterial 16S rRNA gene amplicon PCR primers used were forward primer 341F and reverse primer 805R. After PCR amplification, the PCR products were detected and visualized by 2% (*w*/*v*) agarose gel electrophoresis. The efficient tags were grouped into operational taxonomic units (OTUs) with at least 97% similarity using Usearch software (Version 11.0.667).

### 2.2. Isolation, Purification, and Identification of Spoilage Bacteria

The spoilage bacteria were isolated after modifying the method from Chen et al. [1]. The prepared sample homogenate mixture was serially diluted and 100 μL of the dilution was incubated for 24 h at 25 °C on tryptose soya agar (TSA). Subsequently, individual colonies on agar were isolated and purified based on morphological characteristics such as size, shape, the degree of elevation, edge structure, luster, transparency, texture, and color. Then, they were repeatedly streaked onto the agar plate and Gram staining and microscopy were performed until pure colonies were obtained. The purified isolates were then cultivated in tryptic soy broth (TSB) at 30 °C for 24–36 h, after which 2 mL of TSB culture was centrifuged (10,000× *g* for 5 min) to collect bacteria cells. The total DNA of the bacteria samples was extracted individually using a Bacterial Genomic DNA Extraction Kit (Biomed Biological Technology Co., Ltd., Beijing, China), and 16s rRNA gene fragments (about 1400 bp) were amplified using the primers 27F and 1492R. PCR amplification products were sequenced and identified by the method of Yang et al. [12]. A threshold of 99% similarity was used as preliminary identification for two sequences belonging to the same species. The PCR products were then sent to Sangon Biotech (Shanghai) Co., Ltd. (Shanghai, China). for sequencing. The sequencing results were compared using the BLAST (version 2.14.0, https://blast.ncbi.nlm.nih.gov/Blast.cgi, accessed on 16 December 2024), and the selected dominant strains were stored in TSB with 25% sterilized glycerol at −80 °C for later use.

### 2.3. Sterilized Treatment and Inoculation of Oyster Meat

Shucked oysters were thoroughly cleaned with running potable water to remove contaminants, including mud, sand, and fouling organisms. The washed oyster meat was processed to make sterile flesh according to Zhuang et al. [13].

Three distinct strains from the dominant bacterial species were selected to inoculate the oyster meat. The three dominant spoilage bacteria, *Ps. immobilis*, *S. putrefaciens*, and *P. swingsii,* isolated from spoiled oysters, were used to inoculate the prepared sterile meat blocks, respectively, as mentioned previously. Before inoculation, the selected strains were cultured to a concentration of about 9 Log_10_ CFU/g. Prepared oyster meat samples were then inoculated by immersing them into a bacterial suspension at approximately 6.0 Log_10_ CFU/mL using sterile 0.85% physiological saline.

The first group was immersed in 0.85% sterile saline for 10 min as a control (C) group, and the remaining three groups were immersed in each of the three bacterial suspensions for 10 min. The samples were soaked and then dried for surface moisture in a sterile environment at room temperature, after which they were dispensed in sterile self-sealing bags. The three inoculation groups are denoted as the *Ps. immobilis* (Ps) group, the *S. putrefaciens* (S) group, and the *P. swingsii* (P) group. Each group contained 15 oyster meat samples. For each indicator, three bags were randomly selected for analysis and determination in each group.

### 2.4. Microbiological and Biochemical Analysis of Bacteria-Inoculated Samples

#### 2.4.1. Determination of the Total Viable Count (TVC)

TVC was analyzed using the plate dilution gradient method with slight modifications described by Tan et al. [9]. Five g of the oyster sample was homogenized in 45 mL of sterile saline for 5 min. After 10-fold gradient dilution, 0.1 mL of the suspension was spread on PCA. The TVC was recorded after 48 h of incubation at 37 °C and expressed as Log_10_ CFU/g. Plates with colony numbers between 30 and 300 and no spreading colony growth were selected for counting as the total number of colonies.

#### 2.4.2. Determination of the TVB-N

The TVB-N of the samples was determined according to the semimicrotitration method of Zou et al. [14] with minor modifications. Ultimately, the TVB-N value was established by evaluating the quantity of hydrochloric acid (HCl) utilized during the titration process.

#### 2.4.3. Determination of pH

Ten g of grated oyster meat was taken in a conical flask, 100 mL of deionized water was added, stirred for 30 min, then left to stand and filtered through filter paper to obtain the filtrate for determination using a pH meter (PHS-3, Thunder magnetic instrument Factory, Shanghai, China).

#### 2.4.4. Determination of Thiobarbituric Acid Reactive Substances (TBARSs)

The TBARS value was determined according to the description of Ma et al. [15] with slight modifications. TCA and TBA solutions were prepared at concentrations of 5% and 0.02 mol/L, respectively. Five g of the sample was weighed, 25 mL of prepared trichloroacetic acid (TCA) solution was added, and 1 mL of filtrate was taken after 30 min of ice bath reaction. One mL of the above filtrate and one mL of the prepared thiobarbituric acid (TBA) solution were mixed well, and the reaction was carried out in a water bath at 100 °C for 30 min; the sample was then removed and cooled to room temperature. The absorbance of the final colored solution at 532 nm was then determined using an ultraviolet-visible spectrophotometer (UV-2450, Shimadzu Corporation, Kyoto, Japan).

### 2.5. TCA-Soluble Peptides

The TCA-soluble peptides were determined according to the method of Yang et al. [12]. The tyrosine content of the supernatant was quantified using the method of Lowry et al. [16], and the results were expressed as μmol tyrosine/g.

### 2.6. SDS-PAGE of Oyster Protein

The extraction of myofibrillar protein from oysters and SDS-PAGE were performed as described Zhuang et al. [17] and Zhuang et al. [13]. The protein concentration of the extract was determined using the biuret method and then adjusted to 2.0 mg/mL.

### 2.7. Determination of Weight Loss

Sample weights were obtained before storage (W_1_) and after storage (W_2_), respectively, and the percentage weight loss was calculated according to the following formula [18].(1)$$\mathrm{Weight}\;\mathrm{loss}\;(\%)=\frac{w_{1}-w_{2}}{w_{1}}\times \mathrm{100}$$

### 2.8. LC-MS-Based Untargeted Metabolomics of Inoculated Samples

For polar metabolites, LC-MS/MS analyses were performed using a UHPLC system (Vanquish, Thermo Fisher Scientific, Waltham, MA, USA) with a Waters ACQUITY UPLC BEH Amide (2.1 mm × 50 mm, 1.7 μm) coupled to an Orbitrap Exploris 120 mass spectrometer (Orbitrap MS, Thermo). The oyster meat in each group was individually and fully crushed before sampling. For sampling, 25 mg of a sample was weighed at low temperature, and 500 μL of the extraction solution (methanol: acetonitrile: water = 2:2:1 (*v*/*v*)) was added. The extraction solution contained isotope-labeled internal standards. The above components were vortex-mixed for 30 s. The above mixture was homogenized in a homogenizer (35 Hz, 5 min) and then transferred to an ice-water bath for sonication for 5 min, and this step was repeated three times. Subsequently, the samples were centrifuged at 4 °C (13,800× *g* for 15 min). The supernatant was placed into the injection bottle and tested on the machine. The stability of the instrument was judged by the retention time of the internal standard throughout the assay.

### 2.9. Statistical Analysis

All experiments were conducted three times with three replications. The results are presented as the mean ± standard deviation. Analysis of variance was performed by Duncan’s test using IBM SPSS Statistics 27 (SPSS Inc., Chicago, IL, USA) at a significance level of *p* < 0.05.

## 3. Results and Discussion

### 3.1. Bacteriome Composition of Oyster Meat

The results of 16S rRNA analysis showed the bacteriome composition of oyster meat at the genus level (Figure 1). The dominant bacterial genera identified were *Psychrobacter, Shewanella*, and *Photobacterium*. The relative abundance increased significantly during storage. The relative abundance of *Psychrobacter* increased from 12.82% on day 0 to 58.32% on day 6, and *Photobacterium* increased from 13.24% on day 6 to 61.18% on day 10. *Photobacterium* has been detected as a spoilage bacterium in fish, salmon, and other seafood products in several studies, particularly in high-CO_2_ aeroponic packaging environments [19]. *Psychrobacter* has been found to dominate several types of seafood, including finfish and shellfish, in Greek seawater as well as in the warm and cold waters of other regions [20]. *Psychrobacter* has been identified as a spoilage microorganism for seafood due to its ability to produce metabolites that result in off flavors and the sensory rejection of the product [3,21]. In particular, *Ps. immobilis* hydrolyzes amino acids, which causes some unpleasant fishy odors.

Previous studies have reported that *Shewanella* exhibits high spoilage activity and can cause spoilage in aquatic food products [22]. Under chilled storage conditions, the extracellular hydrolytic enzymes secreted by spoilage bacteria, particularly protein hydrolytic enzymes, are considered the main cause of spoilage in aquatic products [23]. Research has demonstrated that *Shewanella* is the primary spoilage organism in refrigerated Baltic yellowtail and rhubarb fish [24]. *S. putrefaciens* is present in Pacific oysters on the half-shell and on farmed Atlantic cod; it produces trimethylamine and odorous dimethylamine compounds and degrades proteins during storage, leading to lactic acid accumulation [25,26]. According to the analysis, although the initial *Shewanella* content in oyster meat was low, it increased rapidly during storage. *Shewanella* levels increased from 1.11% on day 0 to 6.95% on day 6, indicating a more than fivefold rise. It is worth noting that, of the known spoilage microorganisms in numerous seafood species, *Shewanella* spp. and *Photobacterium* spp. have been detected in Pacific oysters [25]. *Psychrobacter*, *Shewanella*, and *Photobacterium* are all common spoilage bacteria found in seafood. These three rod-shaped, Gram-negative organisms prefer to grow at low temperatures. All three have featured in studies proving that they spoil seafood. Therefore, *Ps. immobilis*, *S. putrefaciens*, and *P. swingsii* were inoculated into sterile oyster meat to test their spoilage potential.

### 3.2. TVC

The growth of microorganisms in oyster meat inoculated with different microorganisms during storage is shown in Figure 2A. The TVC of the control remained below 2 Log_10_ CFU/g throughout the storage period. This result indicates that the control remained relatively fresh and that background microorganisms did not interfere with the experiment. The initial TVCs of the *Ps. immobilis*, *S. putrefaciens*, and *P. swingsii* groups were 5.21, 4.30, and 4.60 Log_10_ CFU/g, respectively. The initial oyster colony number range was met, and there were no significant changes between treatment groups (*p* > 0.05), making it eligible for a study of spoiling capacity. The exponential increase in the bacterial population from day 0 of storage to day 8 indicates that the intervening microorganisms adapted successfully to the oyster meat environment. All three bacteria exhibited rapid growth during storage, exceeding 8 Log_10_ CFU/g by day 10. After 8 days of storage, all the inoculated bacteria entered the stationary phase.

### 3.3. TVB-N

TVB-N is associated with endogenous enzymes and microorganisms, consisting mainly of dimethylamine, ammonia, and trimethylamine [27]. TVB-N is a collective term for the decomposition of proteins by microorganisms and enzymes and the production of nitrogenous compounds with volatile properties. The higher the TVB-N value, the greater the degree of protein breakdown, indicating that the food is less fresh. Fresh oyster meat has an initial TVB-N value of approximately 4.2 mg (100 g)^−1^, as shown in Figure 2B. During the first four days of storage, TVB-N values increased slowly in all samples, particularly in the control group (*p* > 0.05). However, after the sixth day of storage, TVB-N content increased sharply in all inoculated groups. According to Yu et al. [28], a TVB-N value of 15 mg (100 g)^−1^ is an acceptable and appropriate limit for the spoilage of aquatic products. However, the TVB-N concentration in the *Ps. immobilis* and *S. putrefaciens* groups exhibited a considerable increase compared to the *P. swingsii* group, reaching 25.0 mg (100 g)^−1^ at the end of the storage period. This suggests that the samples from these two groups were highly spoiled at the end of storage.

### 3.4. pH Value

After an item of seafood is caught and killed, various complex changes continue to occur within its body, sequentially progressing through three stages: rigor mortis, the resolution of rigor, and spoilage. During the rigor mortis stage, the carbohydrate metabolism pathway in oysters is anaerobic respiration, where glycogen breakdown produces a large amount of lactic acid, and the breakdown of ATP generates a certain amount of phosphoric acid, leading to a decrease in pH. The pH values of the inoculated groups decreased continuously from day 0 to day 6 (Figure 2C). This decrease may be due to the catabolic reactions of carbohydrates, fats, and proteins in oysters after death, with phosphorylase enzymes initiating glycogenolysis and leading to the accumulation of lactic acid, which subsequently lowers the pH of the tissues [4]. As freshness decreased from day 6 to day 10 of storage, the autolysis of the oyster proteins occurred. This led to the decomposition of proteins, amino acids, and other nitrogenous substances into ammonia and amines, raising the pH value. The overall pH values showed a significant downward trend followed by an upward trend, and other researchers have reported similar results for fish filets during refrigeration [28]. Additionally, with the prolonged storage time, bacterial activity gradually increased, with bacteria secreting various enzymes that further accelerated the decomposition of proteins, resulting in a further increase in pH.

### 3.5. TBARS

During food storage, enzymatic hydrolysis and auto-oxidation lead to fat oxidation, resulting in aldehydes and other off-flavor substances. Malondialdehyde can react with TBARS to form a red complex; thus, the TBARS value indicates the extent of fat oxidation. The TBARS of the inoculated group exhibited a higher increase with storage time than that of the control group. TBARS values of 1–2 mg/kg in seawater fish flesh produce undesirable odors and flavors. However, none of the oyster meat in the inoculated groups exceeded these ranges at the spoilage endpoint, likely because oysters are low-fat compared to saltwater fish. According to the results of Figure 2D, it is obvious that on the eighth to tenth day of storage, the TBARS content of the *P. swingsii* group was significantly higher than that of the control group and the other inoculated groups (*p* < 0.05). Therefore, it can be seen that *Photobacterium* spp. has a strong ability to oxidize fat, and this conclusion has been similarly reported before. Lim et al. [29] reported that *Photobacterium* spp. can produce a variety of lipases, and its protein-coding genes encode numerous adhesins, toxins, hemolysins, proteases, and lipases. These findings further illustrate that *P. swingsii* can produce microbial lipase. During storage, enzymatic hydrolysis and auto-oxidation lead to fat oxidation, resulting in ketones, aldehydes, and other off-flavor substances. Therefore, the TBARS content of the *P. swingsii* group was higher than that of the other groups.

### 3.6. Analysis of the Role of Microorganisms in the Hydrolysis of Oyster Proteins

#### 3.6.1. Changes in TCA-Soluble Peptides

As storage time increases, oyster muscle tissue softens and degrades, leading to a decline in quality and a loss of food value. Protein is the primary constituent of muscle tissue, providing structural support and functionality. Therefore, the degradation of protein will directly impact the quality of oysters and serve as a significant indicator of oyster deterioration.

TCA-soluble peptides are generated from the proteolytic breakdown of proteins by endogenous and microbial exogenous enzymes. Changes in their concentration ultimately affect the texture and freshness of seafood. The inoculated groups all showed increased TCA-soluble peptide concentration over time, whereas the control group did not show significant changes (Figure 3A). On the tenth day, the TCA-soluble peptide concentrations in the *Ps. immobilis* and *S. putrefaciens* groups reached 9.51 μmol tyrosine/g and 10.64 μmol tyrosine/g, respectively. *S. putrefaciens* demonstrated higher protein-hydrolyzing activity, resulting in significantly greater TCA-soluble peptide concentrations compared to the C group and the *P. swingsii* group (*p* < 0.05). However, there was no significant difference between the *S. putrefaciens* group and the *Ps. immobilis* group (*p* > 0.05).

#### 3.6.2. Oyster Protein Degradation

SDS-PAGE was utilized to visually characterize the degree of protein degradation. Myofibrillar proteins, including myosin, actin, myogenin, and myoglobulin, significantly influence the texture of aquatic goods by their breakdown and variations in composition. The degradation of protein band around 150 kDa was observed in samples inoculated with *S. putrefaciens*. Additionally, the two bands with a molecular weight of around 48 kDa in the Ps10 and S10 groups were significantly narrower and lighter in color, indicating that *Ps. immobilis* and *S. putrefaciens* actively hydrolyzed these proteins. In contrast, the bands of these two proteins in the Ph10 group showed no signs of thinning or lightening, suggesting that *P. swingsii* has a weak ability to hydrolyze these proteins. Therefore, it can be deduced that the order of the abovementioned spoilage bacterial ability to degrade proteins is *S. putrefaciens* > *Ps. immobilis* > *P. swingsii*.

#### 3.6.3. Weight Loss

The weight loss percentage of the samples is presented in Figure 3C. According to Wang et al. [30], water loss accounts for most of the weight loss in oysters. The water-holding capacity of oyster meat is directly related to and is generally indicated by weight loss. The weight loss percentage of the control group remained consistently below 10% throughout the storage period, while all inoculated groups exhibited a similar reduction pattern. This indicates that the water-holding capacity of the control group oysters during storage is less affected. However, the samples inoculated with *Ps. immobilis* and *S. putrefaciens* showed greater weight loss compared to the samples inoculated with *P. swingsii*. At the end of storage, the weight loss was significantly higher in the *Ps. immobilis* and *S. putrefaciens* groups than in the *P. swingsii* and control groups (*p* < 0.05). This indicates that the water-retention capacity of the oyster muscle is weakened, and its water-holding capacity is decreased. The denaturation and degradation of proteins in seafood affects their ability to hold water. This results in a large amount of water escaping in the form of juice during storage. This further demonstrates that *Ps. immobilis* and *S. putrefaciens* are more capable of degrading proteins than *P. swingsii*.

### 3.7. Metabolomics Analysis

#### 3.7.1. Analysis of Metabolic Metabolite of Oyster Spoilage

Research has been conducted on the ability of spoilage bacteria, including *Psychrobacter* and *Shewanella*, to generate volatile organic compounds in raw, packed, or processed seafood [31]. Based on the results of the above experiments, it can be seen that *P. swingsii* is not as capable of causing oyster spoilage as *Ps. immobilis* and *S. putrefaciens*. Therefore, in the experiments analyzing the metabolomics of spoiled oysters, only the spoiled oysters inoculated with *Ps. immobilis* and *S. putrefaciens* were analyzed, and the experiments related to oysters inoculated with *P. swingsii* were not performed. In this study, 651 metabolites were isolated from the oyster samples and identified utilizing the HMDB database. Among the detected metabolites, organic heterocyclic compounds and organic acids and their derivatives have the highest content, accounting for 20.63% and 11.83%, respectively. The results suggest that changes in oyster quality are closely related to the amounts of metabolites associated with organoheterocyclic compounds (phenylalanine, tyrosine, and tryptophan), which play a role in oyster spoilage. As shown in Figure 4A,B, the horizontal axis indicates the different sample groups, the vertical axis indicates the differential metabolites between the groups. The red color indicates that the substance is highly expressed in the content of the group in which it is present, while the blue color indicates low expression. In the C10 vs. Ps10 comparison, 4-Morpholinopropanesulfonic acid, N-Lactoyl-Phenylalanine, N-Lactoylphenylalanine, Homoarginine, Propionic acid, 7,8-Dimethoxy-3-(4-methoxyphenyl)-4-oxo-4H-chromen-5-yl-2-O-pentopyranosylhexopyranoside, N,N-Dimethylarginine(ADMA), M630T113, 2-(2H-Tetraazol-5-yl)aniline, and 4-Hydroxy-N-desmethyltamoxifen were the differential metabolites. In contrast, in the C10 vs. S10 comparison, 4-(7-Hydroxy-3-(hydroxymethyl)-5-(3-hydroxypropyl)-2,3-dihydro-1-benzofuran-2-yl)-2-methoxyphenyl 6-deoxy-.alpha.-D-mannopyranoside, N-(Naphthalen-2-yl)-2-sulfanylacetamide, 1-Hexadecyl-1H-indole-2,3-dione, Cephalexin, 7,8-Dimethoxy-3-(4-methoxyphenyl)-4-oxo-4H-chromen-5-yl-2-O-pentopyranosylhexopyranoside, (2R)-7-Methoxy-3-oxo-3,4-dihydro-2H-1,4-benzoxazin-2-yl. beta-D-glucopyranoside, Orotic acid, 4-Cyanobenzene-1-sulfonamide, N-Acetylcadaverine, and 1,2,3,6-Tetrahydropyridine-4-carboxylic acid significantly increased differential metabolites. All of the above significantly elevated differential metabolites had the potential to serve as markers of spoilage in chilled stored oysters. As expected, the majority of these compounds were organoheterocyclic compounds and organic oxygen compounds. This corroborated the previous findings that changes in oyster quality may be closely related to the amount of related metabolites, such as organic heterocyclic compounds. Notably, 7,8-Dimethoxy-3-(4-methoxyphenyl)-4-oxo-4H-chromen-5-yl-2-O-pentopyranosylhexopyranoside was significantly increased in the Ps10 and S10 groups after storage, indicating its potential as a spoilage marker for oysters. The spoiled oyster samples contained two special volatiles (trimethylamine and indole). People often use their peculiar, disgusting, and nasty odor as aroma markers to distinguish between fresh and spoiled seafood samples. They are only present when oysters have spoiled, while fresh oysters do not have the characteristic aroma of these two volatiles. However, Figure 4 does not show a significant increase in the content of these two compounds, possibly because GC-MS collected more volatiles than LC-MS. Generally, LC-MS was able to discover higher boiling points and thermally stable volatiles not found by GC-MS. Therefore, among the compounds identified by LC-MS, more are long-chain compounds (typically more than 13 carbons). Additionally, LNT and adenosine 2′-monophosphate were significantly decreased compounds in both C10 and Ps10, suggesting that they have the potential to act as freshness markers for oysters.

The volcano plot provides a clearer perspective on the general distribution of metabolite disparities between groups. In this study, Student’s *t*-test with a *p*-value less than 0.05 and a variable importance in the projection (VIP) greater than 1 were used as criteria to screen for differential metabolites. The results of group C0 versus group Ps10 are shown in Figure 4C. The results of group C0 versus group S10 are displayed in Figure 4D. The top 10 significantly up-regulated and significantly down-regulated differential substances are labeled in the figure. As shown in Figure 4C,D, the significantly elevated differential metabolites in the volcano plot have a clear correspondence with the significantly elevated differential metabolites in the heat map, again indicating that the above metabolites may be able to be used as markers of spoilage.

To further clarify the changes in metabolite content with the freshness of oysters, a correlation analysis was conducted between the significantly increased differential metabolite content in the heatmap and the spoilage indicators of TVC, TVB-N, pH, TBARS, TCA-soluble peptide, and weight loss. As shown in Figure 5, the significantly increased differential metabolite contents in the Ps and S groups of oysters compared to the C10 group are all significantly positively correlated with TVC, TVB-N values, TBARS values, and TCA-soluble peptides contents. Since pH tends to decrease and then increase during oyster storage, there is no strong correlation between pH and other indicators of spoilage as well as metabolites. It is not difficult to find that 1,2,3,6-Tetrahydropyridine-4-carboxylic acid and Propionic acid are the metabolites most closely related to spoilage indicators. Since the correlation analysis includes data from both the Ps group and the S group, it indicates that 1,2,3,6-Tetrahydropyridine-4-carboxylic acid and Propionic acid are strongly associated with the metabolic activities of *Ps. immobilis* and *S. putrefaciens*, which lead to oyster spoilage. There have been reports with similar conclusions to those of this study. Zhang et al. [32] determined that Butanoic acid accounted for 4.69% and Propanoic acid accounted for 2.06% of the top 10 compounds that differed in volatile profile characteristics during oyster storage using the HSSPME method, and he noted that organic acids consisted of the major volatile profile characteristics of oysters during storage. Therefore, 1,2,3,6-Tetrahydropyridine-4-carboxylic acid and Propionic acid have the potential to serve as metabolic markers for oyster spoilage.

#### 3.7.2. Analysis of Metabolic Pathways of Oyster Spoilage

The KEGG pathway analysis of samples inoculated with *Ps. immobilis* and *S. putrefaciens* is shown in Figure 6. Each square in the figure represents a metabolic pathway in the process of oyster meat spoilage. The size of the square indicates the impact factor of the metabolic pathway, and the color of the square indicates the *p*-value of the enrichment analysis (using the negative natural logarithm -ln(p)), with darker colors indicating more significant enrichment. Therefore, the metabolic pathways represented by squares that are larger and darker in color will be the focus of our research. As shown in Figure 6, the squares representing the metabolic pathways for D-glutamine and D-glutamate metabolism are the darkest and largest in both the Ps and S groups, indicating that these pathways are most strongly affected and may serve as potential target pathways for food spoilage [33,34]. In Figure 6, the significantly enriched metabolic pathways include purine metabolism, alanine, aspartate, and glutamate metabolism, and phenylalanine, tyrosine, and tryptophan biosynthesis (-ln *p*-value > 1.5). In the D-glutamine and D-glutamate metabolism pathway, L-glutamic acid and L-glutamine are consumed to produce proline, and then arginine and proline are consumed during metabolism. Syropoulou et al. [35] found that seafood spoilage due to the metabolic activity of cold-tolerant bacteria affected arginine and proline metabolism. Notably, in the entire D-glutamine and D-glutamate metabolism pathways, only L-phenylalanine increased significantly during storage. Given that phenylalanine, tyrosine, and tryptophan biosynthesis are also significantly enriched pathways, L-phenylalanine may be used as a potential spoilage marker in oysters. In this study, inoculating oyster meat with a single strain can eliminate bacterial interactions, thereby facilitating the elucidation of the exact role of the tested bacteria and the analysis of their metabolites. In order to specifically inhibit the process of oyster spoilage and prolong the shelf life in the next step, a preliminary study was carried out. Nonetheless, the true activities of spoilage microorganisms and their potential interactions are also worth exploring through the application of metabolomics in naturally spoiled oyster meat.

## 4. Conclusions

Overall, three dominant strains of spoilage bacteria were evaluated for their capacity to spoil oyster meat in a 4 °C storage environment. *Ps. immobilis* and *S. putrefaciens* were identified as the most important dominant spoilage bacteria causing the spoilage of oyster meat. Conversely, *P. swingsii* exhibited low spoilage potential during storage but was active in producing microbial lipases that oxidize fats. This study emphasized the crucial impact of the dominant spoilage bacteria in quality changes in oyster meats stored at low temperatures. Additionally, it identified 7,8-Dimethoxy-3-(4-methoxyphenyl)-4-oxo-4H-chromen-5-yl-2-O-pentopyranosylhexopyranoside, 1,2,3,6-Tetrahydropyridine-4-carboxylic acid, Propionic acid, and L-Phenylalanine as potential spoilage markers for oysters under cold storage conditions. This study, from the perspective of bacterial spoilage capacity and metabolomics, provides new insights for future research on food spoilage mechanisms. Finally, it is recommended that researchers conduct multi-omics analyses of naturally spoiled oyster meat to study the effects of bacterial interactions on seafood spoilage in the future.

## Figures and Tables

**Figure 1 foods-14-00193-f001:**
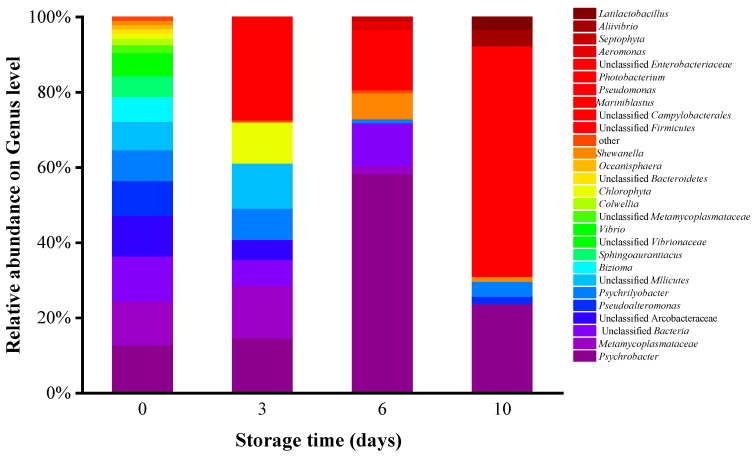
Bacteriome composition in chill-stored shucked oyster meat. (The microbiota composition was calculated at the genus level. Day 0, Day 3, Day 6, and Day 10 represent the storage time of shucked oyster meat.)

**Figure 2 foods-14-00193-f002:**
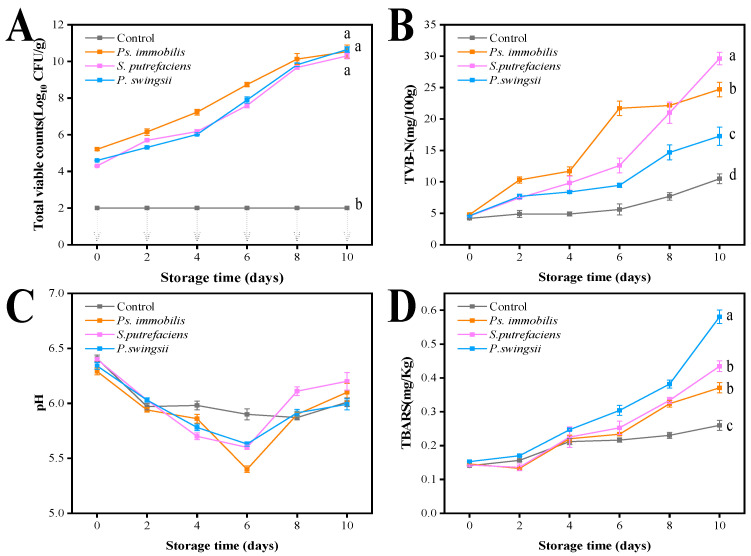
Changes in the biochemical indices of spoilage in chill-stored shucked oyster meat inoculated with spoilage bacteria (*Ps. immobilis*, *S. putrefaciens*, and *P. swingsii*). ((**A**): Changes in total viable counts of the control and bacteria-inoculated samples (the downward arrows mean the total viable counts in the control group remained lower than 2.0 Log_10_ CFU/g during the whole storage time). (**B)**: Changes in the TVB-N of the control and bacteria-inoculated samples. (**C**): Changes in the pH of the control and bacteria-inoculated samples. (**D**): Changes in the TBARS of the control and bacteria-inoculated samples.) Different lowercase letters represent significant difference among various groups (*p* < 0.05).

**Figure 3 foods-14-00193-f003:**
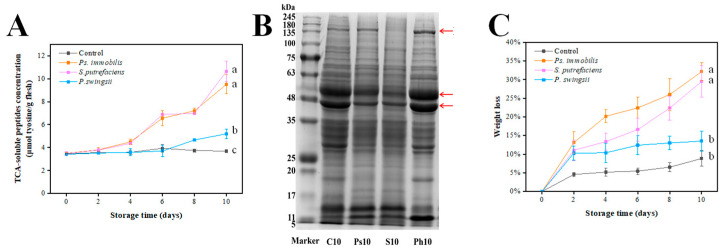
Microbial roles and functions in proteolytical activity. ((**A**): Changes in the TCA-soluble peptides concentration of the control and bacteria-inoculated samples. (**B**): The SDS-PAGE of myofibrillar proteins extracted from the control and bacteria-inoculated samples (the red arrows indicate protein bands with altered concentration. C10, Ps10, S10, and Ph10 represent the control samples, *Ps. Immobilis*-inoculated samples, *S. putrefaciens*-inoculated samples, and *P. swingsii*-inoculated samples on day 10, respectively). (**C**): Changes in the weight loss of the control and bacterial inoculated samples). Different lowercase letters represent significant difference among various groups (*p* < 0.05).

**Figure 4 foods-14-00193-f004:**
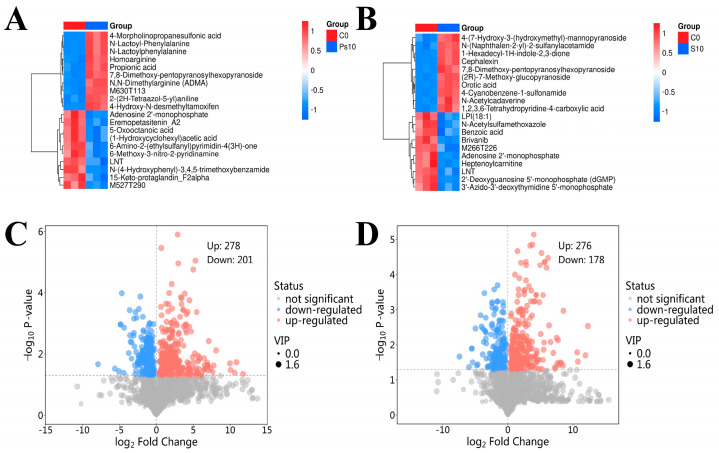
Metabolomics reveals differences in shucked oyster meat metabolites between inoculations with different spoilage bacteria. ((**A**): A heatmap of hierarchical clustering analysis for group C0 vs. Ps10. (Differences between the control and *Ps. Immobilis*-inoculated samples.) (**B**): A heatmap of the hierarchical clustering analysis for group C0 vs. Ps10. (Differences between the control and *S. putrefaciens*-inoculated samples.) (**C**): A volcanogram of differences between the control and *Ps. Immobilis*-inoculated samples. (**D**): A volcanogram of differences between the control and *S. putrefaciens*-inoculated samples. (The top 10 differential substances that were significantly up-regulated and significantly down-regulated were selected in (**C**) and (**D**), respectively)).

**Figure 5 foods-14-00193-f005:**
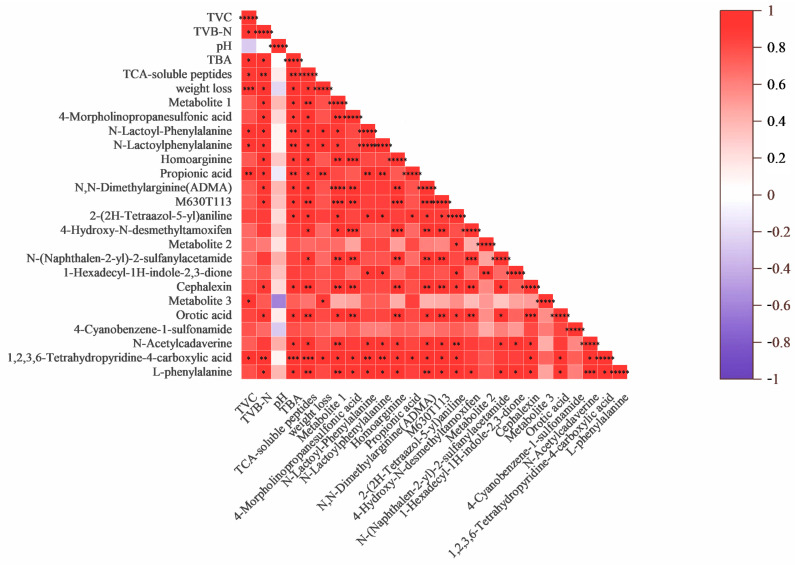
Correlation coefficients between metabolite content and each freshness index (Metabolite 1 represents 7, 8-Dimethoxy-3-(4-methoxyphenyl)-4-oxo-4H-chromen-5-yl-2-O-pentopyranosylhexopyranoside. Metabolite 2 represents 4-(7-Hydroxy-3-(hydroxymethyl)-5-(3-hydroxypropyl)-2,3-dihydro-1-benzofuran-2-yl)-2-methoxyphenyl 6-deoxy-.alpha.-D-mannopyranoside. Metabolite 3 represents (2R)-7-Methoxy-3-oxo-3,4-dihydro-2H-1,4-benzoxazin-2-yl. beta-D-glucopyranoside. (*: *p* <= 0.05, **: *p* <= 0.01, *** *p* <= 0.001, **** *p* <= 0.0001, ***** *p* <= 0.00001)).

**Figure 6 foods-14-00193-f006:**
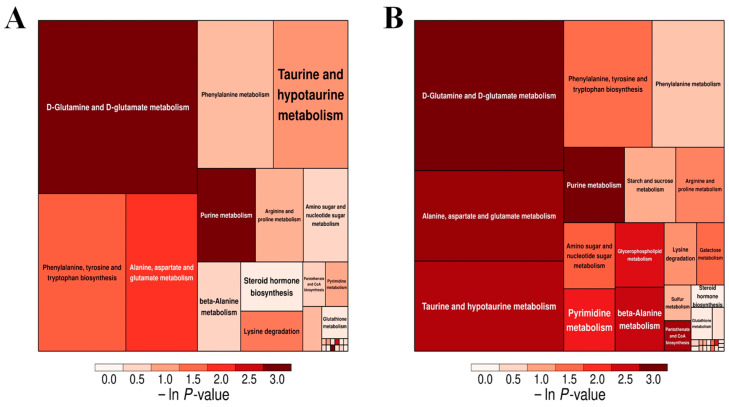
(**A**): Pathway analysis for group C0 vs. Ps10. (Differences between the control and *Ps. Immobilis*-inoculated samples.) (**B**): Pathway analysis for group C0 vs. S10. (Differences between the control and *S. putrefaciens*-inoculated samples).

## Data Availability

The original contributions presented in this study are included in the article. Further inquiries can be directed to the corresponding author.

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
