# Peer review of "Studies on Quality Deterioration and Metabolomic Changes in Oysters Induced by Spoilage Bacteria During Chilled Storage"

_foods, 2025, doi:10.3390/foods14020193_

Round 1

Reviewer 1 Report

Comments and Suggestions for Authors

The manuscript Foods-3404970 entitled “Studies on quality deterioration and metabolomic changes of oysters induced by spoilage bacteria during chilled storage reports the research on the bacteriome of degradation of oysters and the impact of degradation bacteria at biochemical level. The study shows scientific quality and can have impact on strategies to improve the quality of seafood. Nevertheless, the manuscript requires changes to achieve the required quality in the Foods Journal.

Below are the recommended changes.

Line 24. Delete the extra t

Line 25-26. This statement does not add any value to the study; if it is preliminary a proper study is required.

Line 31. modify: the same word mentioned twice in a raw is not appropriate

Line 35. Change to and tend

Line 37. These particular spoilage organisms? what are they?

Line 48. change to: by bacteria through glycolytic pathway and fermentation.

Line 90-91. this sentence is not clear: rewrite.

Line 95 PAGE purified: what this means?

Line 110. How the Bacterial DNA was extracted? information required.

Line 112-113 This sentence previously to the information about sequencing does not make sense. Modify.

Line 116. what culture medium was used?

Line 125 change 9 log CFU/g to 9 Log10 CFU/g. Here and all over the manuscript use Log10

Line 126 there is no bacterial solution; bacterial cells do not dissolve! change to bacterial suspension!

Line 126-127. the concentration of physiological saline is required.

Line 139 gradient dilution… what gradient? decimal?

Line 143 TVB-N- what this means?

Line 151 indicate the proper information about the equipment used.

Line 152 TBARS- in full first

Line 154 TCA and TBA - the same as above in full first

Line 167 change to was performed as described

Line 175 change inoculated bacteria samples to inoculated samples

Line 194 Change to Bacteriome: The authors sequenced the 16 rRNA gene. 

Microbiota is the set of the microbial cells of a specific environment (constituted by bacteria, fungi protozoa, algae and virus); Microbiome is the genomic content of the microbiota.

Line 196 The same as above

Line 199 The same as above

Line 202 change rose to increased. The same in line 223.

Line 224 change increase to raise

Line 225 spp. is not in italic.

Line 237 statistical significance between the variables must be shown in the graphs.

Line 246 change microbial to bacterial

Line 294 indicate the significance difference.

Line 315-316. again indicate the significant differences.

Line 317-328 Where are these results illustrated? How we know the identity of these proteins in a SDS-PAGE?

Line 336 change to bacterial

Line 345 how significantly higher?

Line 406 eliminate speaking

Line 422-427 the identification of the parameters at the x-axis requires improvement, the reader is not able to identify them.

Line 509 the citation of the references through the text must comply with the rules of the Journal

Author Response

  1. Line 24. Delete the extra t

Answer: We have deleted it as suggested.

  1. Line 25-26. This statement does not add any value to the study; if it is preliminary a proper study is required.

Answer: The sentence has been deleted.

  1. Line 31. modify: the same word mentioned twice in a raw is not appropriate.

Answer: “Worldwide, people” has been changed to “People around the world”.

  1. Line 35. Change to and tend

Answer: Revisions have been made in the manuscript.

  1. Line 37. These particular spoilage organisms? what are they?

Answer: These particular spoilage organisms include Shewanella putrefaciens, Pseudomonas spp., Photobacterium spp., Aeromonas spp., etc. Revisions have been made in the manuscript.

  1. Line 48. change to: by bacteria through glycolytic pathway and fermentation.

Answer: Revisions have been made in the manuscript.

  1. Line 90-91. this sentence is not clear: rewrite.

Answer: We have rewritten it in the manuscript.

  1. Line 95 PAGE purified: what this means?

Answer: This is one type of primers of high purity provided by the company. After the primers are synthesized, they are usually purified in order to ensure the purity and quality of the primers. PAGE purification is one of the commonly used techniques. This process purifies PCR primers by separating them according to the size of the DNA fragments through polyacrylamide gel electrophoresis, removing impurities that may have arisen during synthesis, such as unreacted raw materials, dimers, and so on.

  1. Line 110. How the Bacterial DNA was extracted? information required.

Answer: Total DNA of bacteria samples was extracted individually via Bacterial Genomic DNA Extraction Kit (Biomed Biological Technology Co., Ltd., Beijing, China).

  1. Line 112-113 This sentence previously to the information about sequencing does not make sense. Modify.

Answer: Thanks. Here 16 rRNA gene fragments were amplified and then sequenced to confirm the identification of the specific bacteria. We have revised it to make it clearer.

  1. Line 116. what culture medium was used?

Answer: TSB with 25 % glycerol was used here.We have added the information in the revised manuscript.

  1. Line 125 change 9 log CFU/g to 9 Log10 CFU/g. Here and all over the manuscript use Log10

Answer: Revisions have been made in the manuscript.

  1. Line 126 there is no bacterial solution; bacterial cells do not dissolve! change to bacterial suspension!

Answer: Revisions have been made in the manuscript.

  1. Line 126-127. the concentration of physiological saline is required.

Answer: We have added the concentration as suggested in the revised manuscript.

  1. Line 139 gradient dilutionwhat gradient? decimal?

Answer: A 10-fold gradient dilution was used in the manuscript, and has been revised in the manuscript.

  1. Line 143 TVB-N- what this means?

Answer: TVB-N is total volatile basic nitrogen, which has been explained in the revised manuscript.

  1. Line 151 indicate the proper information about the equipment used.

Answer: Equipment information has been added in the revised manuscript.

  1. Line 152 TBARS- in full first

Answer: Full name of TBARS has been added in the revised manuscript.

  1. Line 154 TCA and TBA - the same as above in full first

Answer: Full names of TCA and TBA have been added in the revised manuscript.

  1. Line 167 change to was performed as described

Answer: Revisions have been made in the manuscript as suggested..

  1. Line 175 change inoculated bacteria samples to inoculated samples

Answer: Revisions have been made in the manuscript as suggested...

  1. Line 194 Change to Bacteriome: The authors sequenced the 16 rRNA gene.

Microbiota is the set of the microbial cells of a specific environment (constituted by bacteria, fungi protozoa, algae and virus); Microbiome is the genomic content of the microbiota.

Answer: We have changed it to bacteriome as suggested.

  1. Line 196 The same as above

Answer: We have changed it to bacteriome as suggested.

  1. Line 199 The same as above

Answer: Revisions have been made in the manuscript as suggested.

  1. Line 202 change rose to increased. The same in line 223.

Answer: Revisions have been made in the manuscript as suggested.

  1. Line 224 change increase to raise

Answer: Revisions have been made in the manuscript as suggested.

  1. Line 225 spp. is not in italic.

Answer: We have italicize it as suggested.

  1. Line 237 statistical significance between the variables must be shown in the graphs.

Answer: We have added the significance symbols in the revised figures.

  1. Line 246 change microbial to bacterial

Answer: Revisions have been made in the manuscript.

  1. Line 294 indicate the significance difference.

Answer: Revisions have been made in the manuscript.

  1. Line 315-316. again indicate the significant differences.

Answer: Revisions have been made in the manuscript.

  1. Line 317-328 Where are these results illustrated? How we know the identity of these proteins in a SDS-PAGE?

Answer: Thanks for the comment. The results of TCA-soluble peptides was shown in Fig 3A. We have added specific figure number after description of the results in the revised manuscript. For the protein bands in SDS-PAGE, we previously predict the identity of bands based on size and previous literature. But we agree with the reviewer that it is not accurate to identify a band only from molecular weight. Therefore ,we have deleted the protein name of the band (such as MHC) and only mentioned them as protein bands around specific molecular weight in the revised manuscript.

  1. Line 336 change to bacterial

Answer: Revisions have been made in the manuscript.

  1. Line 345 how significantly higher?

Answer: Thanks. We have added p < 0.05 in the revised manuscript.

  1. Line 406 eliminate speaking

Answer: Revisions have been made in the manuscript.

  1. Line 422-427 the identification of the parameters at the x-axis requires

improvement, the reader is not able to identify them.

Answer: We have adjusted the identification of the parameters at the x-axis to make it clearer now.

  37. Line 509 the citation of the references through the text must comply with the rules of the Journal

Answer: The format of the references has been adjusted as required by the journal.

Reviewer 2 Report

Comments and Suggestions for Authors

In this work, the role of three dominant spoilage bacteria strains in oyster spoilage is explored. It is an interesting work; it provides information that may be of the interest for the scientific community.

The manuscript is well organized and written. Only some issues need to be adressed for a better comprehension.

1.      Separate the units from the numbers; as an example, change 4℃ for 4 ℃.

2.      In section 3.2, the sentence “After 8 days of storage, all the inoculated bacteria entered the stationary pase” should be supported.

3.      The initial TVCs of the Ps. immobilis, S. putrefaciens, and P. swingsii groups were 5.21, 4.30, and 4.60 log CFU/g, respectively; why different values were used?

4.      Figure 3 should be improved.

5.      In the abstract it was stated that the study offers valuable insights for the development of innovative preservation techniques; it is required information for supporting this aseveration. 

Author Response

  1. Separate the units from the numbers; as an example, change 4℃ for 4 ℃.

Answer: Revisions have been made in the manuscript.

  1. In section 3.2, the sentence “After 8 days of storage, all the inoculated bacteria entered the stationary phase” should be supported.

Answer: Thanks for pointing this out. We agree with the reviewer that this statement is not justified, therefore we have deleted in the revised manuscript.

  1. The initial TVCs of the Ps. immobilis, S. putrefaciens, and P. swingsii groups were 5.21, 4.30, and 4.60 log CFU/g, respectively; why different values were used?

Answer: Samples were immersed in three bacterial suspensions at the same concentration of approximately 6.0 Log10 CFU/mL, but it is difficult to keep the initial TVC exactly the same for different samples, which is commonly seen in similar studies (Li, 2020).

Li, Y., Jia, S., Hong, H., Zhang, L., Zhuang, S., Sun, X., Liu, X., & Luo, Y. (2020). Assessment of bacterial contributions to the biochemical changes of chill-stored blunt snout bream (Megalobrama amblycephala) fillets: Protein degradation and volatile organic compounds accumulation. Food Microbiology, 91, 103495. https://doi.org/10.1016/j.fm.2020.103495.

  1. Figure 3 should be improved.

Answer: We have improved the quality of Fig3 as suggested.

  1. In the abstract it was stated that the study offers valuable insights for the development of innovative preservation techniques; it is required information for supporting this aseveration.

Answer: Thanks. We have revised the sentence as follows:  These findings extend our understanding of the roles that microorganisms play in the spoilage of oysters and offers valuable insights into the development of technologies for monitoring freshness of oysters based on those potential spoilage markers. 
